



# Origin of the Bohai Sea, North China Craton and implication for bi-directional back-arc extension in East Asia continental margin

Alan Liu Chen [1], Xuanhua Chen [2]

[1] Northview High School, Johns Creek, GA 30097, USA

[2] SinoProbe Laboratory, Chinese Academy of Geological Sciences, Beijing 100037, China

*Correspondence to*: Xuanhua Chen (xhchen@cags.ac.cn)

**Abstract.** The Bohai Sea in eastern China is located in the back-arc extensional regime due to westward subduction of the Pacific Plate underneath the Eurasian Plate. It is one of the regions with frequent earthquakes. Previous recognition of the origin of the Bohai Sea was limited by the understanding of back-arc extensional mode perpendicular to the subduction zone

in eastern Asian continental margin. In this paper, a new model for the genesis of the Bohai Sea is proposed, based on the construction of major fault system and investigation of several main boundaries enclosing the Bohai Sea. Through field investigation and analyses of tectonic landforms and boundary faults on the northwest coast of the Bohai Sea and eastern and western margins of the Liaodong Peninsula, and geological correlation of the Liaodong and Jiaodong Peninsulas and surrounding areas, we revealed a left-lateral strike-slip fault between the northwest coast of the Liaodong Bay and western

margin of the Liaodong Peninsula, and proposed a right-lateral strike-slip fault between the eastern margin of the Liaodong Peninsula and northwestern margin of the Jiaodong Peninsula. This mode of movement may have been resulted from the NE stretching which is parallel to the subduction zone in northwestern Pacific margin. Therefore, we suggest that the formation of the Bohai Sea is resulted from the superimposition of the NE extension parallel to the subduction zone on the NW extension perpendicular to the subduction zone. We speculate that the two-direction extension perpendicular and parallel to

the subduction zone should be the basic pattern of the back-arc extension with spherical-geometric effect.

## 1 Introduction

The Bohai Sea is located in eastern part of the North China Craton (NCC; Fig. 1; Chen et al., 2022a). Traditionally, the Bohai Sea is included in the Bohai Bay Basin (BBB). The BBB consists of the North China Plain (also known as the North China Basin), Bohai Sea, and Lower Liaohe Plain, with an area of ca. 200,000 km² (Hou et al., 1998). It is always considered

as an intracontinental extensional rift basin in back-arc setting, resulted from the westward subduction of the Pacific Plate underneath the Eurasian Plate in late Mesozoic and Cenozoic (Allen et al., 1997; Liang et al., 2016). It is also one of the regions with strong earthquake activities in East Asia (Xiao et al., 2004; Yin et al., 2015; Chen et al., 2020; Fig. 2). The remarkable feature of the BBB is the thinned crust and lithosphere with high geothermal gradient, due to craton deconstruction and mantle uplifting (Guo et al., 2005; Li et al., 2012; Zhu and Xu, 2019; Zhu et al., 2020; Zhou et al., 2022).



Although the water depth in the Bohai Sea is only an average of 18 meters, however, the Cenozoic sediments are widely

distributed in this area, with a total thickness of over 8000 meters, making them as an important source rock series of oil and

gas (Xiao et al., 2004).



**Figure 1: Simplified tectonic map of Asia, showing the location of the Bohai Sea in eastern China. Modified from Chen et al. (2022a). KOM, Kolyma-Omolon superterrane (Ar-P). BJ, Bureya-Jiamusi superterrane (Ar-J). TLF, the Tan-Lu fault. Locations of Figs. 2A and 3 are also shown.**



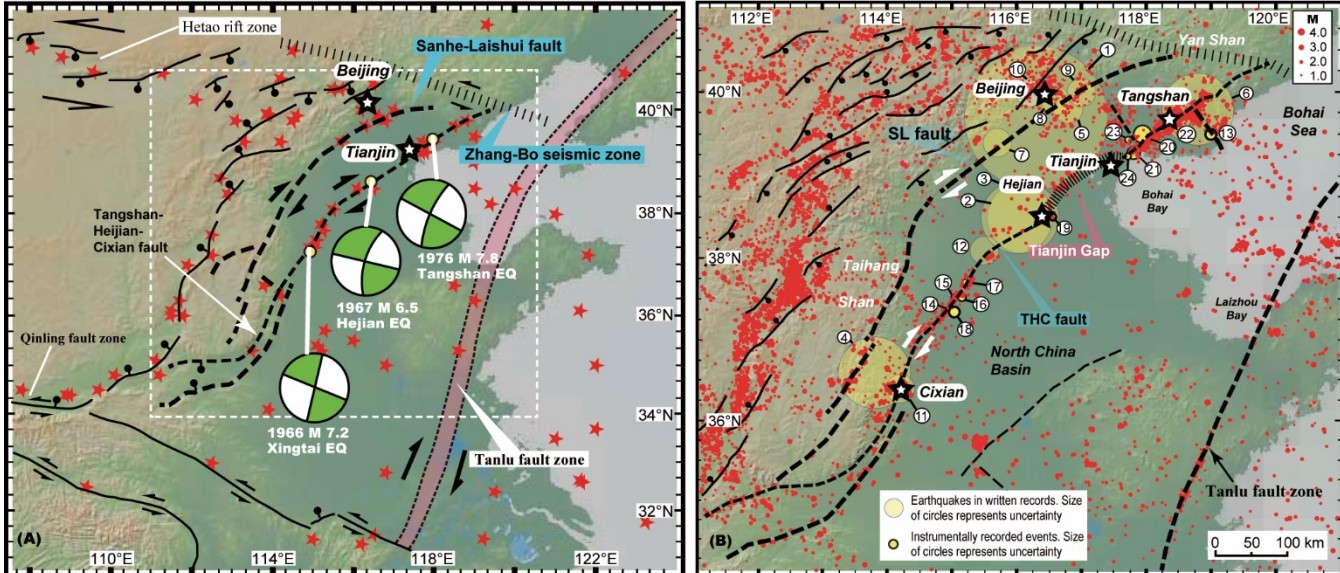

**Figure 2: (A)** Pre-instrumentation historical earthquakes (EQ) with M ≥ 6 across north China and focal mechanisms of the A.D. 1966 Xingtai, 1967 Hejian, and 1976 Tangshan earthquakes (mainly right-lateral strike-slipping along the THC; after Yin et al., 2015). **(B)** M≥6 earthquakes from A.D. 1000 to the present in the North China Basin against a background of microseismicities between 2009 and 2013 (after Yin et al., 2015). THC—Tangshan-Hejian-Cixian fault; SL—Sanhe-Lushui fault. For earthquakes that occurred in the same year, they are labelled sequentially, such as 1966-1, 1966-2, 1976-1, and 1976-2. Earthquakes: (1) A.D. 1057 S. Beijing Earthquake (M 6.8); (2) A.D. 1068 Hejian Earthquake I (M 6.5); (3) A.D. 1144 Hejian Earthquake II (M 6.0); (4) A.D. 1314 Shexian Earthquake (M 6.0); (5) A.D. 1536 Tongxian Earthquake (M 6.0); (6) A.D. 1624 Luanxian Earthquake (M 7.0); (7) A.D. 1658 Laishui Earthquake (M 6.0); (8) A.D. 1665 W.Tongxian Earthquake (M 6.5); (9) A.D. 1679 Sanhe Earthquake (M 8.0); (10) A.D. 1730 W. Beijing Earthquake (M 6.5); (11) A.D. 1830 Cixian Earthquake (M 7.5); (12) A.D. 1882 Shenxian Earthquake (M 6.0); (13) A.D. 1945 Luanhe Earthquake (M 6.3); (14) A.D. 1966-1 Xingtai Earthquake (M 6.8); (15) A.D. 1966-2 Xingtai Earthquake (M 6.7); (16) A.D. 1966-3 Xingtai Earthquake (M 7.2); (17) A.D. 1966-4 Xingtai Earthquake (M 6.2); (18) A.D. 1966-5 Xingtai Earthquake (M 6.0); (19) A.D. 1967 Hejian Earthquake (M 6.5); (20) A.D. 1976-1 Tangshan Earthquake (M 7.8); (21) A.D. 1976-2 Changli Earthquake (M 6.2); (22) A.D. 1976-3 Luanxian Earthquake (M 7.4); (23) A.D. 1976-4 Ninghe Earthquake (M 6.9); (24) A.D. 1977 Tanggu Earthquake (M 6.2) (from Yin et al., 2015).

Regarding to the origin of the Bohai Sea and BBB, there is still significant controversy. The main viewpoints proposed by previous studies are: 1) The BBB is a back-arc intraplate rift basin with lithospheric extension (Guo et al., 2005; Li et al., 2012; Liu et al., 2018; Zhou et al., 2022); 2) a pull-apart basin resulted from right-lateral strike slipping along the Tan-Lu and Taihang Shan faults due to subduction of the Pacific Plate in Cenozoic (Hou et al., 1998; Hu et al., 2022; Liu and Wu, 2022), as a part of the right-lateral pull-apart basin system in NW Pacific region (Xu et al., 2014); 3) a result of active mantle plume with a diameter of ca. 600-800 km (Xiao et al., 2004); 4) superimposed effect of multiple-phase extensions and strike-slip deformations (Allen et al., 1997; Liu and Wu, 2022). The formation and evolution of the BBB reflects superimposed effects of multiple episodes of back-arc extensional and strike-slip deformation (Liu and Wu, 2022). Historically, the BBB area has experienced many strong earthquakes, including the 1597 M>7, 1679 M8.0 Sanhe, 1830 M7.5 Cixian, 1888 M7.5, 1966 M7.2 Xingtai, 1969 M7.4 Bohai Bay, 1975 M7.3 Haicheng, 1976 M7.8 Tangshan, and 1976 M7.4 Luanxian earthquakes (Fig. 2; Deng et al., 1976; Yin et al., 2015; Chen et al., 2020). Present GPS velocity field and focal mechanism solution of





the 1975 M7.3 Haicheng earthquake showed NNW-SSE stretching stress in the Liaodong Bay and surrounding area (Deng et al., 1976; Wang et al., 2014; Zhao et al., 2015).

Some unresolved key scientific issues on genesis of the Bohai Sea and BBB are listed as follows: 1) Whether the Tan-Lu fault passes through the Bohai Sea, or how it passes through the Bohai Sea; 2) What is the mechanism for the Bohai Sea formed? In this paper, we propose a new model of two-direction back-arc extensional origin for the Bohai Sea, based on detailed analyses on boundary geometry and fault system in the Bohai Sea and BBB, as well as geological correlation analyses of the Jiaodong and Liaodong peninsulas and surrounding area (Fig. 3).

## 2 Regional geological background

### 2.1 Geological overview of the Jiaodong and Liaodong Peninsulas and Jidong Terranes

The Bohai Sea is surrounded by the Jiaodong Peninsula in southeast, the Liaodong Peninsula in northeast, the Jidong Block and Yanshan orogenic belt in northwest, the North China Plain in south, and the Liaohewan Plain in north (Fig. 3). Structural relationship between the Jiaodong and Liaodong peninsulas is the key issue to solving geological problems of the Bohai Sea.

The Jiaodong Peninsula, also known as the Jiaodong Block, is located in the central part of eastern coast of China. It is mainly composed of three major tectonic units: the Jiaobei Terrane in the north, Jiaolai Basin in the middle, and Sulu Orogen in the south (Fig. 3). The Jiaobei Terrane is the most southern part of the NCC, mainly composed of Archean TTG rocks, i.e., tonalites, trondhjemites, and granodiorites, gneisses, such as biotite gneisses and plagioclase amphibolites, and Archean to Paleozoic metamorphic rocks. Intrusive rocks include Triassic granites (225-200 Ma; Koua et al., 2022), Jurassic composite Linglong pluton (170-145 Ma; Yang et al., 2017), and two-stage Early Cretaceous granites (130-126 Ma and 121-116 Ma; Koua et al., 2022; Dong et al., 2023). The Jiaobei Terrane experienced rapid exhumation in 120-95 Ma (Zhang et al., 2022a), with development of extensional structures, such as the Linglong extensional dome (Fig. 3; Zhu et al., 2020; Yan et al., 2021), as well as supersized Jiaodong-type or decratonization-type gold deposits in late Early Cretaceous (Deng et al., 2020; Zhu et al., 2020, 2024; Yang et al., 2021; Zhang et al., 2022a). The Sulu Orogen is located on the southeast side of the Wulian-Qingdao-Yantai fault zone (WQYF), characterized by the occurrence of high to ultra-high pressure metamorphic rocks. It is an Indosinian collisional suture zone between the NCC and South China block (Zhu et al., 2020; Dong et al., 2023). The Jiaolai Basin is located between the Jiaobei Terrane and Sulu Orogen, as a graben basin formed in late Early Cretaceous. It has a high elevation ≥2.0 km in Late Cretaceous (ca. 80 Ma), which was a part of the coast mountains on the eastern margin of the Asian continent (Zhang et al., 2016). Cenozoic basalts outcrop in the Penglai area, eastern Jiaodong Peninsula (Fig. 3).

The Liaodong Peninsula, also known as the Liaodong Block, is located in the northeast of the NCC. It is bordered by the North Yellow Sea fault (NYSF) with the North Yellow Sea Basin (Tian et al., 2007). It is mainly composed of Archean TTG rocks, Paleoproterozoic Liaohe Group metamorphic rocks, Mesoproterozoic to Paleozoic metamorphic sedimentary rocks, and Mesozoic to Cenozoic sedimentary and magmatic rocks. It is dominated by a large number of granites with ages







**Figure 3: Structural geological map of the Bohai Sea and surrounding region (modified from Allen et al., 1997, Ren et al., 2013, Yin et al., 2015, Kim et al., 2018; Zhai et al., 2019; Yang et al., 2020; Ren et al., 2023, and Zhu et al., 2024), showing distribution of fault system and location of field observation sits (see Figs. 4, 5, 6, 7, 8, 9). The Liaonan metamorphic core complex and Linglong dome are from Zhu et al., 2020, Yan et al., 2021, and Ren et al., 2023. Diamondiferous kimberlites are from Liu et al., 2019. JDB, the Jidong block. JBT, the Jiaobei terrane. JB, the Jiaolai basin. SLO, the Sulu orogen. MCC, metamorphic core complex. WZMCC, the Waziyu (also named Yiwulvshan) metamorphic core complex (Sun et al., 2022). KLD, the Kalaqin Dome (Yang et al., 2020). HF, the Honglazi fault. EBF, the East Bohai fault. JLF, the Jiao-Liao fault as the boundary faults of the Jiaodong and Liaodong peninsulas predicted in this study. NYSF, the North Yellow Sea fault (Tian et al., 2007). YYF, the Yilan-Yitong fault. DMF, the Dun-Mi fault. YF, the Yalvjiang fault. WQYF, the Wulian-Qingdao-Yantai fault. JXF, the Jiashan-Xiangshui fault. Strata systems: Q, the Quaternary. N, the Neogene. K, the Cretaceous. Nh, the Nanhuan. Qb, the Qingbaikou. Pt$_3$, the Upper Proterozoic. Jx, the Jixian. Pt$_1$, the Lower Proterozoic. Ar, the Archean. βQ, Quaternary basalts. Granitoids: γK, Cretaceous. γJ, Jurassic. γT, Triassic. γPt, Proterozoic. γAr, Archean.**

of Triassic (231-200 Ma), Jurassic (183-152 Ma), and Cretaceous (139-117 Ma) (Fig. 3; Yan et al., 2021; Zeng et al., 2022; Zhu et al., 2024). It has experienced Yanshanian intracontinental compressional deformation initiated at ca. 171 Ma in Middle Jurassic (Ren et al., 2023), with a mature continental arc formed in Late Jurassic due to the Paleo-Pacific subduction (Zeng et al., 2022). Granitoids plutons, such as the Shizhuzi magmatic complex, intruded with ages ranging from 130 to 126 Ma, indicating asthenosphere upwelling-related craton destruction in Early Cretaceous (Wu et al., 2021; Yang et al., 2021; Wang et al., 2022). Simultaneously, extensional structures, such as the Liaonan metamorphic core complex, developed in late Early Cretaceous (Fig. 3; Zhu et al., 2020; Yan et al., 2021; Ren et al., 2023), accompanied by the occurrence of Cu, Mo, and decratonization-type gold deposits (Wu et al., 2021; Yan et al., 2021; Yang et al., 2021; Zhu et al., 2024). Typical gold deposits in the Wulong-Sidaogou and Xinfang regions have metallogenic ages of ca.120 Ma (Zhang et al., 2022b).

The Jidong Block and Yanshan orogenic belt is located in northern NCC and northwest coast of the Bohai Sea (Fig. 3). The Archean basement rocks outcropped here are mainly consisting of gray gneisses and TTG rocks, as well as supracrustal rock series in granulite facies metamorphism (Zhu et al., 2020). The Jidong Block has experienced the development of Late Paleoproterozoic to Mesoproterozoic Yanliao Rift System (Zhu et al., 2020), and Yanshanian intracontinental orogeny in late Middle Jurassic to early Early Cretaceous (Dong et al., 2015; Yang et al., 2020). Post orogenic extension occurred in late Early Cretaceous (135-100 Ma), represented by the Yunmengshan and Yiwulvshan metamorphic core complexes, as well as the Kalaqin and Fangshan extensional domes (Liu et al., 2017; Yang et al., 2020; Zhu et al., 2020; Sun et al., 2022).

The Jiaobei Terrane, Liaodong Peninsula, and Jidong Block are all part of the NCC, composing of Archean metamorphic rocks and Proterozoic greenstone belts. They have suffered similar geological evolution processes in Phanerozoic. They are all located in the back-arc setting of the subducted west Paleo-Pacific Plate, i.e., the Izanagi Plate. They have undergone intracontinental Yanshanian orogeny in late Middle Jurassic to early Early Cretaceous, and extensional faulting in the late Early Cretaceous, with extensive crustal melting in Mesozoic (Dong et al., 2015; Yang et al., 2017; Clinkscales and Kapp, 2019; Zhu et al., 2020; Yan et al., 2021; Chen et al., 2022a; Sun et al., 2022; Dong et al., 2023).

## 2.2 Fault system of the Bohai Sea, North Yellow Sea, and surrounding areas

As a part of the BBB, **the Bohai Sea** is divided into the main sea and three bays, such as the Bohai Bay in the west, Laizhou



Bay in the south, and Liaodong Bay in the north (Fig. 3). It is an important petroliferous basin and one of oil and gas production bases in China, with extremely complex and diverse fault systems. The Liaodong Bay and western North China Plain are dominated by NNE-SSW trending normal faults and extensional right-lateral strike-slip faults, while the Laizhou

Bay and eastern North China Plain is dominated by nearly E-W and NNE-SSW trending normal faults (Fig. 3; Allen et al., 1997; Ren et al., 2002; Li et al., 2012; Hu et al., 2022; Yuan et al., 2022).

Previous studies have suggested two-phase rifting of the BBB in Cenozoic, controlled by the enhanced back-arc extension due to eastward roll-back of the subducted Pacific Plate (Liu et al., 2017; Allen et al., 1997; Hu et al., 2022). The first phase is the development of elongate half grabens in Paleocene to early Eocene, with deposition of the Kongdian and

145 lower Shahejie Formations (Allen et al., 1997). The second is the development of dextral transtensional pull-apart basin in middle Eocene to early Oligocene, with a transition occurred at ca. 45-43 Ma in middle Eocene (Allen et al., 1997; Chen et al., 2022b). Then, the BBB entered the post-rifting development stage in Neogene and Quaternary (Allen et al., 1997; Chen et al., 2022b). The superimposed strike-slipping on extensional stress field controls the formation and evolution of dextral transtensional BBB fault system in Cenozoic (Liu et al., 2018; Hu et al., 2022).

**The North Yellow Sea Basin** is located to the east of the Liaodong Peninsula and northeast of the Jiaodong Peninsula, adjacent to the Bohai Sea (Fig. 3). It has a boundary fault, i.e., the North Yellow Sea Fault in its northwest (Tian et al., 2007). It is characterized by normal faults trending in NE to NNE, ENE to EW, and NW to NNW directions in Neogene and Quaternary (Shen et al., 2013).

## 3 Field observation and structural analyses

### 3.1 Northwest coast of the Bohai Sea and the Liaodong Bay Basin

Near the Honglazi Bay, Huludao City, the northwestern coast of the Liaodong Bay outcrops the Neoarchean gneisses, Mesoproterozoic Changcheng system, and Lower Triassic Hongla Formation ($T_1h$; Li et al., 2020), intruded by Jurassic and Early Cretaceous granitoids (Fig. 3; Ren et al., 2013). They form coast hills and cliffs with high relieves (Fig. 4A). The Hongla Formation is composed of purplish red cobble and sandy conglomerates, sandy mudstones, and sandstones from the

160 bottom to the top. It is characterized by the development of cross bedding of sandstones and siltstones (Li et al., 2020).

Field observations show several greyish-green mafic dicks intruded in clastic rock series of the Hongla Formation, and a series of thrusts, extensional normal faults, and strike-slip faults developed on the outcrop (Figs. 4B and 4C). According to the regional late Mesozoic extension event, we interpret the mafic dicks as formed during the lithosphere extension in late Early Cretaceous. The dicks are nearly parallel to the normal faults with en echelon arrangement (such as the $F_1$ and $F_2$ in

Fig. 4C). These normal faults and dicks are truncated by the normal faults formed in later stage (such as the $F_3$), and then cut by a detachment fault (the $F_4$) that developed at the bottom of the conglomerate layer in much later stage (Fig. 4C). This detachment fault (the $F_4$) is interpreted as a result of intensive regional extension in late Early Cretaceous, similar to the gravitational collapse in post-orogenic stage. The conglomerate layer above the detachment fault has suffered conformal









**Figure 4: Structural analyses in northwestern coast of the Bohai Sea. (A) The Honglazi Bay in northwestern Bohai Sea, with formation of the Lower Cretaceous cut by coastal cliff formed in Cenozoic. View to south. (B) The Honglazi fault (HF) with flower structure in western coast of the Bohai Sea, formed in the Early Cretaceous. The flower structure is truncated by normal fault system. $T_1h$, Hongla Formation of the Lower Triassic. (C) Development of the extensional normal fault system with intrusion of mafic dikes later than the HF. (D) The western half thrust system of the flower structure related to the HF. (E) Duplex in the Lower Cretaceous related to the flower structure of the Honglazi fault.**

folding, with its front tip (the western wing) inversed (Fig. 4C), indicating the thrusting in detachment front. A thin layer of fault gouge is also observed on the detachment plane.

The Honglazi fault (HF in Figs. 3, 4B and 4C) strikes in N30°E, with nearly vertical faulting plane. Thrusts such as the $F_6$, $F_7$, and $F_8$ faults in Figs. 4B and 4E, constitutes a flower structure related to the Honglazi fault, indicating the latter has the attribute of left-lateral strike slipping. In the outcrop scale, the occurrence of a stacked anticline in the Hongla Formation indicates imbricate thrusting in the fault system (Fig. 4E), implying also left-lateral slipping along the Honglazi fault. Early developed thrust faults, such as the $F_6$ and $F_7$, have cut through the much earlier developed normal faults, such as the $F_9$, and then were cut by the later normal faults, such as the $F_3$ and $F_4$ (Figs. 4C and 4D).

Structural analyses revealed the following deformational and magmatic sequence in the Honglazi area: 1) Regional extension and normal faulting, with mafic dicks intrusion in the Lower Triassic Hongla Formation along extensional fractures, in late time of Early Cretaceous; 2) Left-lateral strike slipping along the Honglazi fault, accompanied by imbricate thrusts, flower structure, and stacked anticline, in early time of Late Cretaceous; 3) Normal faulting along the Honglazi fault in early Cenozoic. Resulting from the continuous extensional faulting and strike slipping in Cenozoic, the Liaodong Bay area continued to receive fine-grained clay and siltstone sedimentation in the Liaohe River Delta, and subsided to form the Liaodong Bay Basin and therein abundance of wetlands (Fig. 5).

### 3.2 Northwest coast of the Liaodong Peninsula

In the Beizuizi area, northeast coast of the Liaodong Bay, late Early Cretaceous granitoids, with an age of 127-120 Ma (Wang et al., 2023), intruded into the Paleoproterozoic Gaixian Group sequence (Fig. 3). A strike-slip fault developed in the granitoids, with a strike of N49°E and dip angle of 76°. We named it as the Beizuizi fault, which could be a branch of the East Bohai fault (EBF in Fig. 3). Early-stage joints indicate the left-lateral strike slipping along the Beizuizi fault (Fig. 6), which is consistent with the movement direction of the Honglazi fault in northwest coast of the Liaodong Bay. The fault truncated the two-stage joints in the granitoids (Fig. 6), implying the faulting later than ca. 120 Ma. Orientated arrangement of potassium feldspars in the granitoids forms general stretching lineation with an orientation of 100° and dip angle of 3°, indicating early-stage horizontal left-lateral strike slipping along the Beizuizi fault post-Early Cretaceous. Fine-grained quartz veins developed on the fault plane, with occurrence of bookshelf structures obliquely to the fault, implying transtensional dextral movement along the fault in later relaxation stage. The arrangement of late-stage joints also reflects the right-lateral transtensional faulting along the Beizuizi fault (Fig. 6). This kind of right-lateral transtensional activity could be



inferred in early Cenozoic, which is consist with the widespread right-lateral strike-slip faulting in the Liaodong Bay and

surrounding area in early Cenozoic (Fig. 3; Allen et al., 1997; Hu et al., 2022).

**Figure 5: The Red Beach wetland located in the Liaohe Estuary area, Liaodong Bay, northern Bohai Sea. The red color is coming from red plants called Suaeda salsa (seepweed), which is a type of grass suitable for survival in saline alkali soil.**

Horizonal left-lateral strike-slip faulting is also found in the Jiangjunshi area, central of northwest coast of the Liaodong Peninsula. The Qingbaikou Formation (Qb) of the Neoproterozoic in this area is mainly composed of pure white quartz sandstones in medium to coarse grained, with nearly horizontal bedding (Fig. 7A). Multiple sets of joints are developed perpendicular to the bedding of the Qingbaikou Formation. Striations on joint surface indicate left-lateral movement in direction of N53°E (Fig. 7B), consistent with the movement direction of the Beizuizi and Honglazi faults. Therefore, we

speculate that there was also Late Cretaceous to early Cenozoic left-lateral strike-slip faulting in southeast coast of the Liaodong Bay, which could be named as the East Bohai fault (Fig. 3).





**Figure 6: The Beizuizi fault, a branch of the East Bohai fault, occurs in northwestern coast of the Liaodong Peninsula, showing a left-lateral strike-slip movement similar to that on the Honglazi fault in Fig.4. Both the Beizuizi and Honglazi faults are considered as relic faults formed before the open of the Bohai Sea.**

### 3.3. Southeast coast of the Liaodong Peninsula

Neogene red layer, with nearly horizontal bedding, outcrops in the Dajiao area, Dalian City on southeast coast of the Liaodong Peninsula. It is a kind of residual deposits in paleo-crust of weathering, mainly composed of magenta clay layer with a thickness of ca. one meter. It is covered by Quaternary clastic deposits in parallel unconformity, and underlain by folded dark gray shales of the Early-Middle Jurassic Wafangdian Formation (Fig. 8). The Wafangdian Formation is a set of fluvial to lacustrine facies coal-bearing strata, mainly composed of sandstones, mudstones, and dark gray shales, interbedded with mudstones. Significant tectonic relief between the coastal zone and the Cenozoic sediments in the Yellow Sea, implies the existence of an active normal fault at coastal cliff. It may be a branch of the North Yellow Sea fault (NYSF in Fig. 3). As a reasonable inference, the southwestward extending of the North Yellow Sea fault should form the Jiao-Liao fault that separates the Jiaodong Peninsula from the Liaodong Peninsula (Fig. 3).



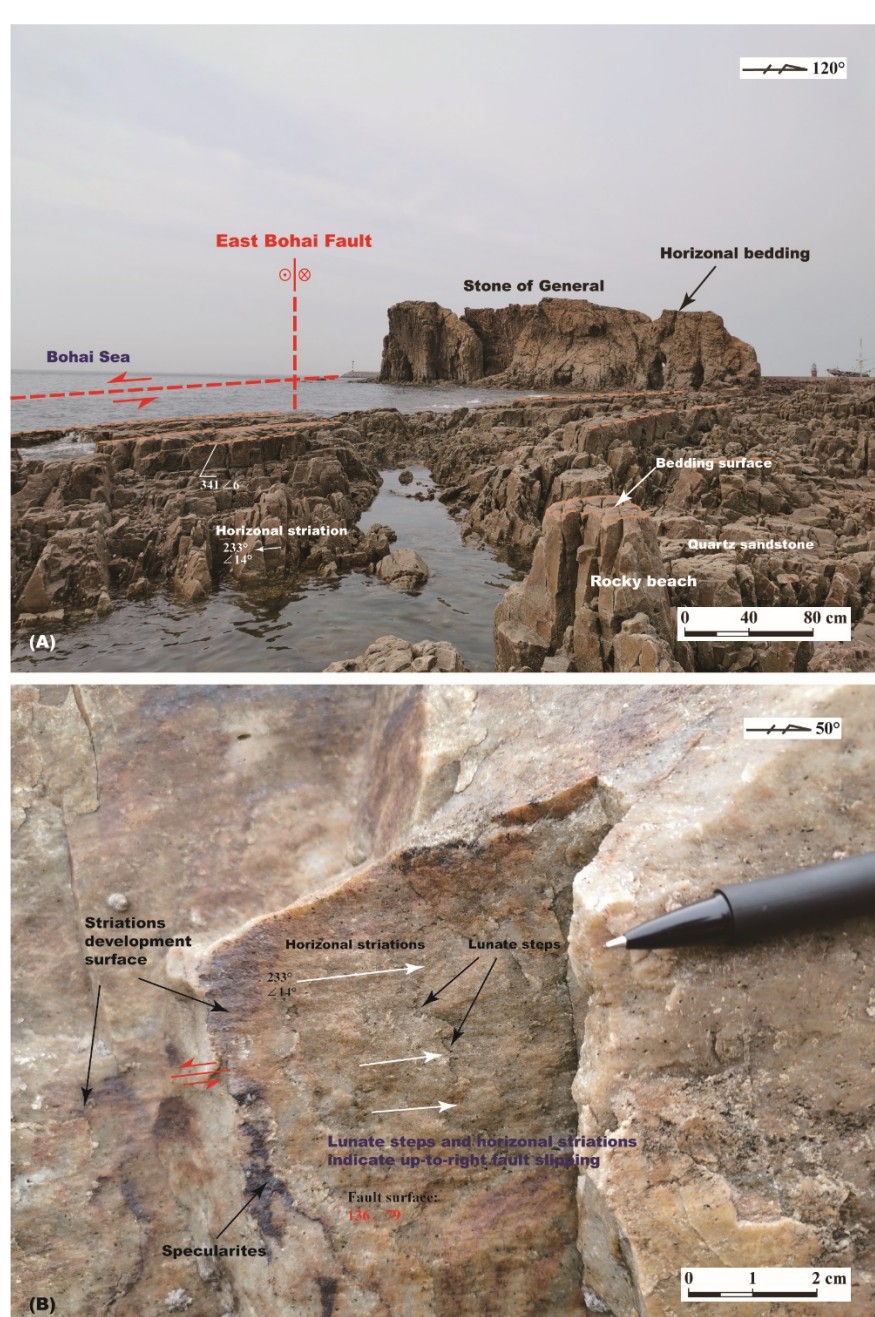

**Figure 7: (A) Topographic feature in western coast of the Liaodong Peninsula, showing geometric properties of the East Bohai fault. (B) Near horizontal striations indicate early stage left-lateral strike-slipping along the East Bohai fault in western coast of the Liaodong Peninsula.**


Active normal faulting also appears in the Laotieshan area, southwest corner of the Liaodong Peninsula. In this area, the significant tectonic relief occurs between the Neoproterozoic Nanhua System and offshore deposits in the North Yellow Sea,



expressed by the coastal landform such as the Elephant Trunk Hill (Fig. 9). The Nanhua System here is composed of pure white medium coarse grained quartz sandstone with thin layers of meta-argillaceous siltstones.


**Figure 8: Coastal cliff occurs at the southernmost edge in eastern coast of the Liaodong Peninsula, showing the vertical cutting of the horizontal Neogene red bed due to active normal faulting. $Q_{1-2}w$, Lower-Middle Jurassic Wafangdian Formation.**

## 4 Geological correlation and proposal of tectonic model

### 4.1 Tectonic relationship among the Jiaodong and Liaodong Peninsulas and Jidong Block


The Liaodong Peninsula and Jiaobei Terrane are both parts of the NCC. They are commonly referred to as the Jiao-Liao Block or a part of the Jiao-Liao-Jilin tectonic belt (Zhu et al., 2020). However, there is still some controversy over the way in which the two peninsulas are connected. Most researchers believe that both the Jiaodong and Liaodong Peninsulas are located on the eastern side of the Tan-Lu fault zone (Figs. 2 and 10A; Xu et al., 1987; Allen et al., 1997; Wang et al., 2000;



Zhu et al., 2004; Li et al., 2012; Clinkscales and Kapp, 2019; Zhu et al., 2020, 2024; Yan et al., 2021; Chen et al., 2022a; Chen et al., 2022b; Hu et al., 2022; Zhou et al., 2022; Ren et al., 2023). Additionally, such a configuration does not really resolve the problem that how they are interconnected. Nevertheless, if we take the early Cretaceous granitoids and extensional structures in consideration, the situation will be greatly improved.

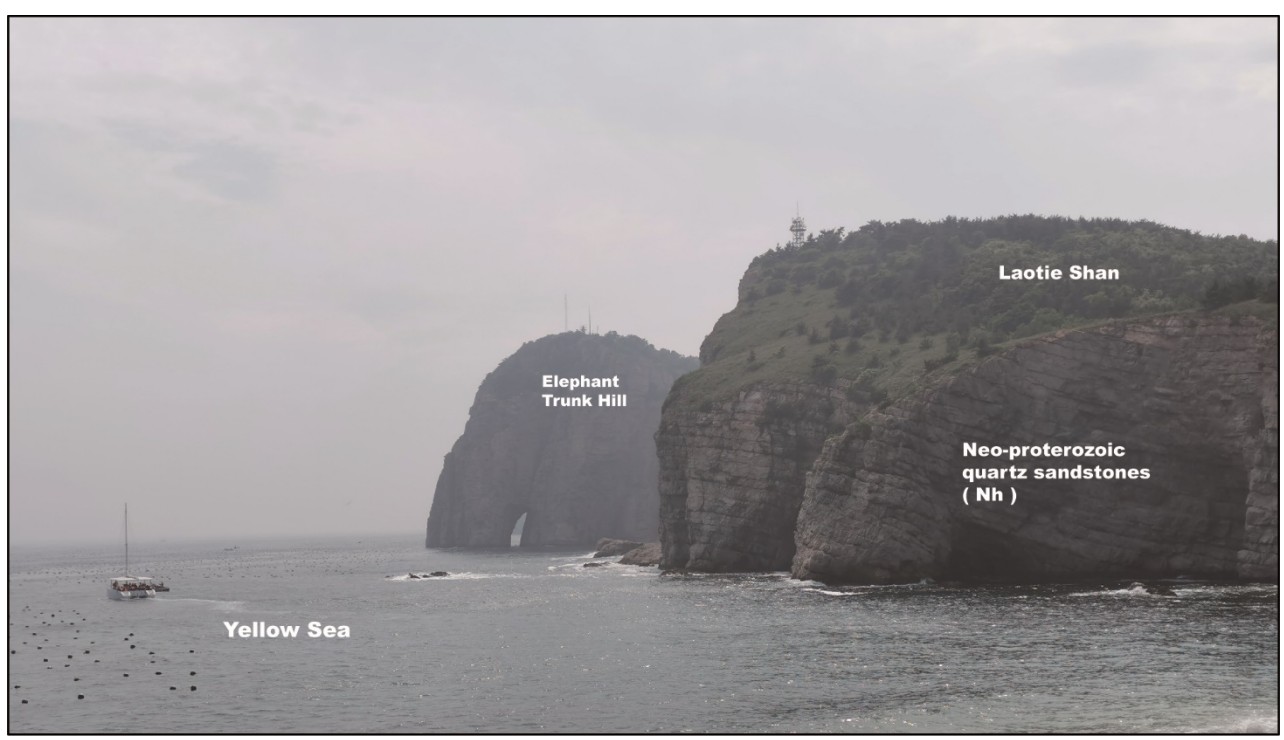


**Figure 9: Great topographic reliefs (cliffs) in southernmost edge of the Liaodong Peninsula, which are interpreted as results from active normal faulting. Nh, the Nanhuan System. Cape of Fishing, Dalian. View to 235°.**

        Early Cretaceous granitic plutons and extensional structures such as metamorphic core complexes or extensional domes
occur in both the Liaodong and Jiaodong Peninsulas (Zhu et al., 2020, 2024; Wu et al., 2021; Yan et al., 2021; Ren et al., 2023). They are interrelated, just like the case in North America (Zuza et al., 2022). They are the two critical control factors related to the Jiaodong- or decratonization-type gold mineralization in late Early Cretaceous (125-115 Ma; Yang et al., 2021). Therefore, we can take the Early Cretaceous granitoids, extensional structures, and gold deposits as piercing points, to reconstruct the spatial relationship between the two peninsulas in Early Cretaceous. Our reconstruction is shown in Fig. 10B,
which predicts the existence of a right-lateral strike-slip fault, referred as the Jiao-Liao fault, between the Jiaodong and Liaodong Peninsulas (Figs. 3 and 10B). Through the recovery of strike-slipping along the Jiao-Liao fault, the Liaonan metamorphic core complex in the Liaodong Peninsula can be jointed with the Linglong dome in the Jiaodong Peninsula (Fig. 10B). Meanwhile, the gold deposit cluster in the Wulong-Sidaogou area of the Liaodong Peninsula can also be buckled up on



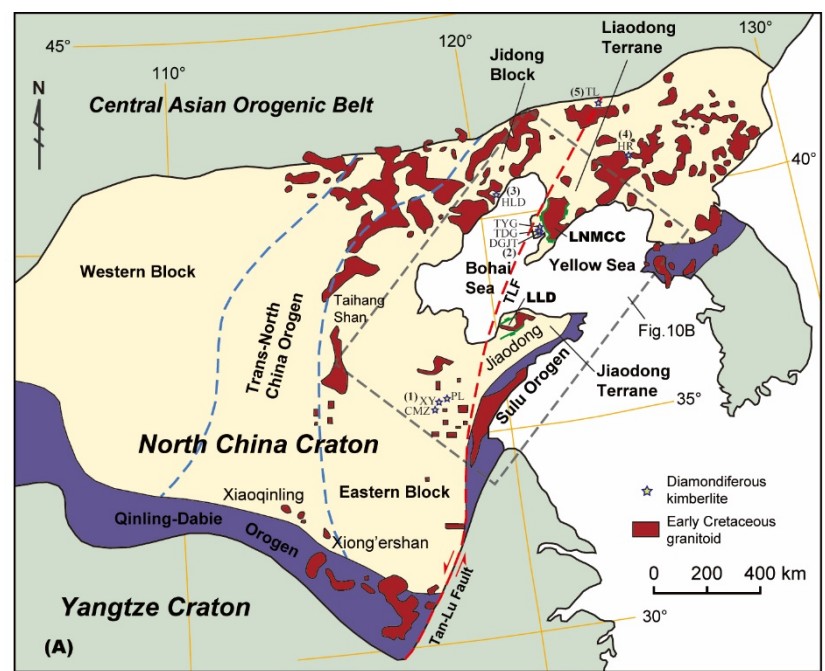

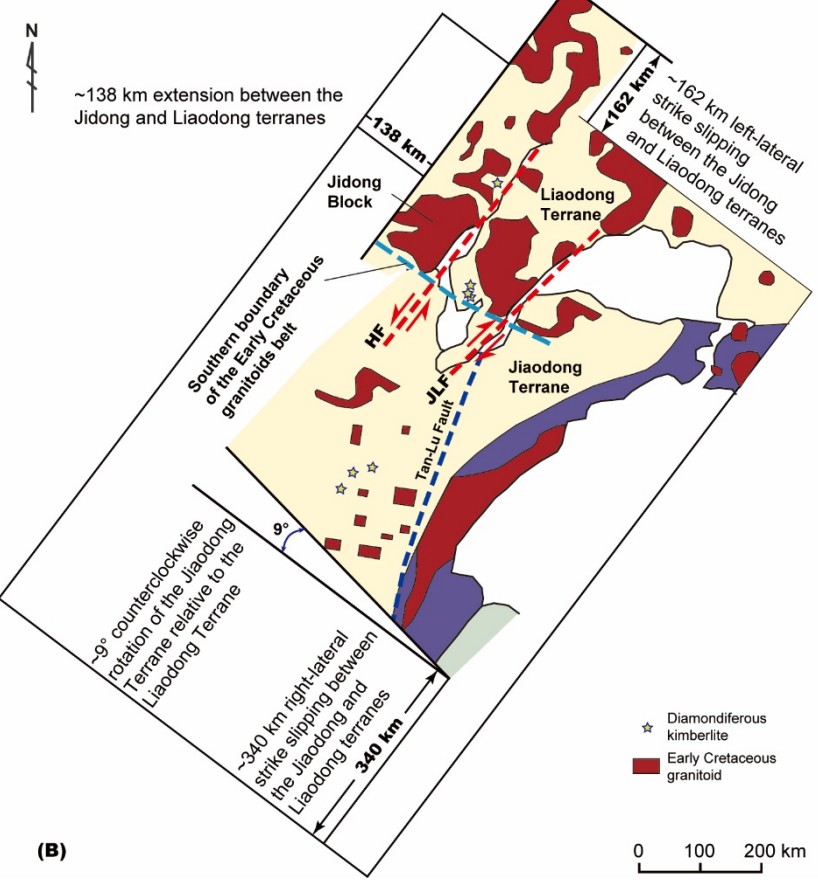





**Figure 10:** Correlation of the Jidong (Yanshan), Jiaodong and Liaodong terranes, according to the distribution of Early Cretaceous magmatic intrusions and diamondiferous kimberlites. (A) Distribution of Early Cretaceous granitoids and diamondiferous kimberlites in the Jiaodong and Liaodong peninsulas (modified from Liu et al., 2019 and Wu et al., 2021). HF, the Honglazi fault. JLF, the Jiao-Liao fault. TLF, the Tan-Lu fault predicted by previous researches. LLD, the Linglong dome. LNMCC, the Liaonan metamorphic core complex. Diamondiferous kimberlites: (1) Mengyin area: PL, Poli; XY, Xiyu; CMZ, Changma Zhuang. (2) Wafangdian area: TYG, Taiyang Gou; TDG, Toudao Gou; DGJT, Dagaojia Tun. (3) HLD, Huludao. (4) HR, Huanren. (5) TL, Tieling. (B) Reconstruction of the Ji-Lu-Jiao-Liao Terrane, regarding to the restoration of Early Cretaceous magmatic complex.

that in the Linglong-Jiaojia area of the Jiaodong Peninsula. The northeastward extending of the Jiao-Liao fault may be connected to the Yalvjiang fault (Fig. 3).

The Jidong and Liaodong Blocks have similar geological compositions and tectonic evolution histories. They share a common tectonic history in Mesozoic, with the typical Yanshanian intracontinental orogeny in Late Jurassic to early Early Cretaceous, and significant craton destruction and extensional faulting in late Early Cretaceous (Dong et al., 2015; Yang et al., 2020). They have approximately simultaneously granitic intrusion events in Early Cretaceous. With an assumption of nearly east-west zonal distribution of the Early Cretaceous plutons, our restoration shows that the Liaodong Bay has opened through the NW-SE extension in late Mesozoic and early Cenozoic, and modified by the left-lateral strike-slipping along the Honglazi and/or East Bohai faults (Fig. 10B). Before the opening of the Liaodong Bay, these two faults should be branches of the same major fault, which could be inferred as the Honglazi fault.

## 4.2 Tectonic relationship between the Jiaodong and Korean Peninsulas

The Jiaobei Terrane and northern Korean Peninsula are both components of the NCC, with the outcrop of Archean TTG metamorphic rocks (Zhai et al., 2019). They are characterized by the Early Cretaceous extensional structures with strikes in NE and extension in NW (Dong et al., 2015), as well as the Cenozoic normal faulting with strike in WNW. They are both suffered from basalt eruption in Quaternary (Fig. 3). This is to say, they have a common pre-Cenozoic history in geological evolution, and the same tectonic setting in Cenozoic. Previous correlation between the Sulu orogen in the south of the Wulian-Qingdao-Yantai fault and Gyeonggi Massif in northern Korea, indicates they are both belong to the high- and ultra-high pressure metamorphic belt formed in Indosinian North and South China collision (Fig. 3; Li et al., 2012; Kim et al., 2018). Therefore, the Jiaodong Peninsula is not simply connected to the Liaodong Peninsula in NNE direction, but a wedge-like connection with both the Liaodong and Korean Peninsulas (Figs. 3, 10, and 11). Restoration of tectonic processes, such as strike slip and normal faulting in an extensional setting, is necessary.

## 4.3 A genetic model of the Bohai Sea based on tectonic correlation

According to the correlational analyses of the Jiaodong, Liaodong, and Jidong Blocks, combined with field observations and structural analyses, we proposed a three-stage kinematic model for the formation and evolution of the Bohai Sea (Fig. 11). Stage 1, in early period of Early Cretaceous, strike slip faulting initiated among the Jidong, Liaodong, and Jiaodong Blocks,



parallel to the paleo-subduction zone, forming the Jiao-Liao, East Bohai, and Honglazi faults (Fig. 11A). Stage 2, in late

Early Cretaceous, as a result of the roll-back of subducting Pacific Plate, extensive back-arc extension occurred at the continental margin of East Asia. The extension deformation is expressed in two directions, i.e., parallel and perpendicular to the subduction zone. The proto-Bohai Sea formed in this stage, as the combined result of the extension and accompanying strike slip faulting (Fig. 11B). Stage 3, the present-day Bohai Sea formed as a result of the continuous bi-directional back-arc extension and strike-slipping along the Jiao-Liao, East Bohai, and Honglazi faults in Cenozoic (Fig. 11C).

310

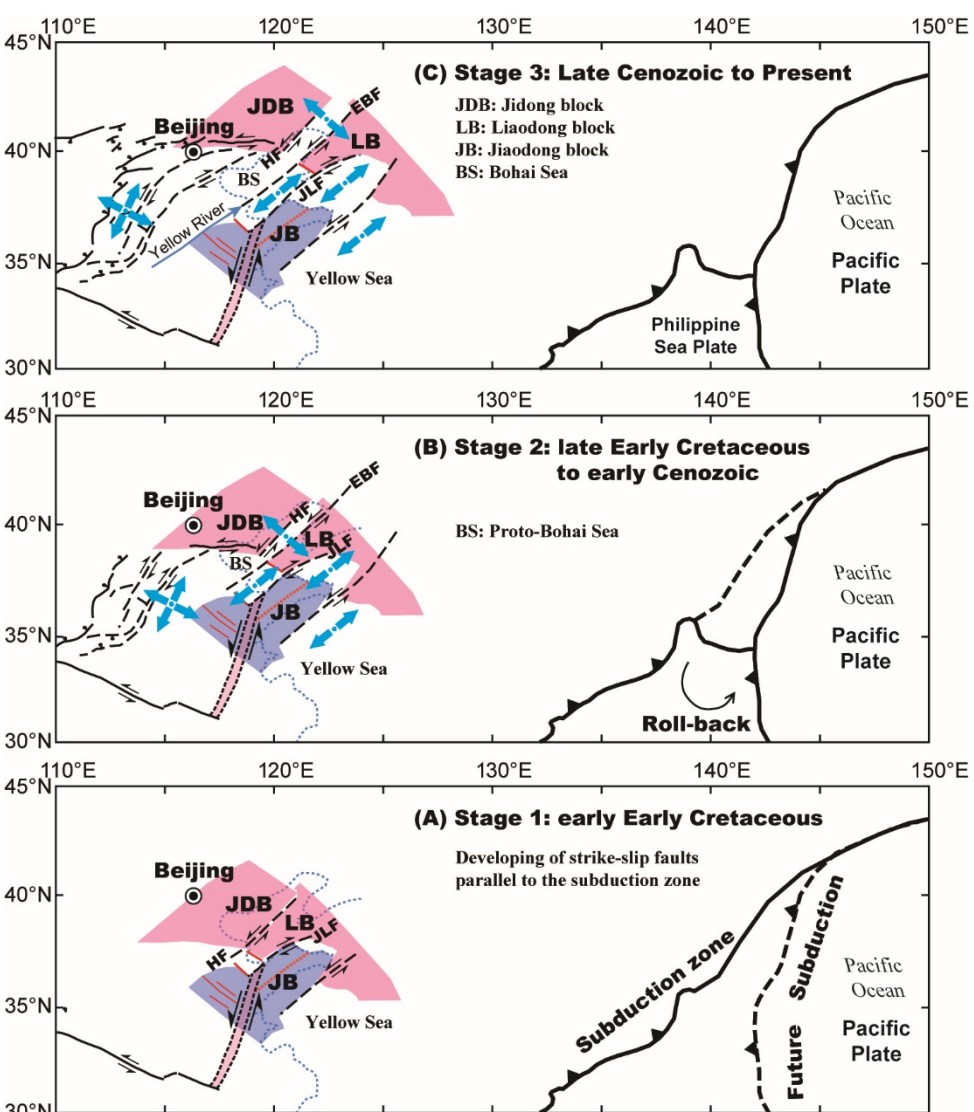

**Figure 11: Proposed three-stage model for the formation of the Bohai Sea as a result of complex faulting in Bohai Sea area and roll-back of the West Pacific plate subduction. JDB, the Jidong Block. LB, the Liaodong Block. JB, the Jiaodong Block. BS, the Bohai Sea.**



## 5 Tectonic significances of the genetic model

### 5.1 Reconstruction of the fault system and re-recognition of the Tan-Lu fault

Some researchers believe that the formation of the Bohai Sea is mainly controlled by the northeast-striking left-lateral strike-slipping Tan-Lu fault zone (Zhu et al., 2004; Min et al., 2013; Zhang et al., 2015). The Tan-Lu fault zone is considered as a major active fault zones in eastern China, starting from the Lujiang City in Anhui Province, with a total length of ca. 2400 km (Xu et al., 1987; Wang et al., 2000; Zhu et al., 2004; Min et al., 2013; Zhang et al., 2015; Zhu et al., 2020). It is divided into several segments, such as the south segment in Anhui and Jiangsu provinces, the Shandong segment (the Yishu Fault Zone; Fig. 3), the Bohai Sea segment from the Weifang City in Shandong Province to the Shenyang City in Liaoning Province (Zhang et al., 2015), and the northeast segment in northeast China, with a total left-lateral displacement of 1000-1500 km (Xu et al., 1987). It is considered as the eastern boundary of the Bohai Sea and the BBB (Hou et al., 1998; Zhou et al., 2022). The main extensional structures in the Bohai Sea are considered to be derivatives of the Tan-Lu fault (Hou et al., 1998), or resulted from the dextral transpression of the pre-existing large-scale NNE strike-slipping fault (i.e., the Tan-Lu fault) in the basement (Xiao et al., 2004). Some other researchers believe that the Tan-Lu fault can be divided into two segments, the south and the north, with the Bohai Sea in the middle. These two segments have different faulting histories, and formed the single Tan-Lu fault in Late Jurassic due to opposite growth of the faults (Li et al., 2023).

The Tan-Lu fault zone is characterized by a large-scale sinistral strike-slip faulting (Xu et al., 1987), especially in its southern segment (Liu et al., 2017). It truncated the Hong'an-Dabie and Sulu high- and ultra-high pressure metamorphic belts, with a sinistral displacement of ca. 540 km (Leech and Webb, 2013). It possibly initiated during the collision between the North China and Yangtze blocks in Triassic (244-209 Ma; Chen et al., 2000), and suffered from counterclockwise rotation of the Lower Yangtze Block, east side of the Tan-Lu fault, in Jurassic (189-164 Ma; Chen et al., 2000; Wang et al., 2000). In late Early and early Late Cretaceous (130-94 Ma), the Tan-Lu fault zone extended northwards into the Yishu fault zone, a rift zone in Luxi area (Fig. 3; Chen et al., 2000). In this time, both the Luxi and Jiaodong regions are characterized by normal faulting, implying a close connection of the extension with the NCC destruction (Li et al., 2018; Zhu and Xu, 2019; Zhu et al., 2020, 2024). Newly achieved paleomagnetic study yields sinistral slip of ca. 100 km, along the southern segment of the Tan-Lu fault in Anhui and Jiangsu provinces, during early Late Cretaceous (100-80 Ma; Qin et al., 2022).

However, there are still some controversies over whether the Bohai Sea segment is a part of the Tan-Lu fault zone (Zhang et al., 2015), and whether the Tan-Lu fault is connected with the Yilan-Yitong fault and/or Dunmi fault to the north (Min et al., 2013). For example, a large number of nearly east-west trending normal faults, as well as some NE-trending normal faults and right-lateral strike-slip faults, formed in the Dongying Depression, southwestern Laizhou Bay, in Cenozoic (Yuan et al., 2022). That is to say, there is no direct evidence for the continuous NE-wards extending of the Tan-Lu and/or Yishu fault zone in Cenozoic. If the Tan-Lu fault zone be designated as a large-scale left-lateral strike-slip fault in East Asia (Chen et al., 2000), the Cenozoic normal faulting and right-lateral strike slipping activities in the Bohai Sea and surrounding




area (Allen et al., 1997; Chen et al., 2022b; Hu et al., 2022; Yuan et al., 2022), should be excluded in the scope of the Tan-Lu and/or Yishu fault zone.

The Luxi area has experienced multi-stage extensional faulting in NE striking, at ca. 61 Ma, 49-42 Ma, and 36-32 Ma, respectively (Li et al., 2018). The extensional direction is parallel to the strike of the Honglazi, East Bohai, and Jiao-Liao faults, implying close connection between the normal faulting and strike-slip movement in Cenozoic. Universally developed right-lateral strike slip faulting in the Bohai Sea area since middle Eocene, should be tightly connected with the right-lateral slipping along the Jiao-Liao fault. The northern segment of the Jiao-Liao fault is the North Yellow Sea fault (NYSF in Fig. 3). It deeply cuts through the lower crust, and extends northeastward to be connected with the Yalvjiang fault (Tian et al.,
2007).

## 5.2 Tectonic reconstruction of the Bohai Sea area

Based on our reconstruction of the fault system and genetic model of the Bohai Sea and surrounding area (Fig. 11), as well as the distribution of Early Cretaceous granites, we got some estimations of the displacement magnitudes among several blocks around the Bohai Sea (Fig. 10B). Among them, the Jiaodong and Liaodong blocks are connected through the Jiao-Liao fault,
with a right-lateral displacement of ca. 340 km. Meanwhile, the Jiaodong Block may have undergone a counterclockwise rotation of ca. 9°, relative to the Liaodong Block. The displacement between the Jidong Block (or Yanshan Orogenic Belt) and Liaodong Block can be partitioned into left-lateral displacement of ca. 162 km along the Honglazi fault, and stretching displacement of ca. 138 km perpendicular to the strike-slip fault (Fig. 10B). Our model does not need to consider the influence of the Tan-Lu fault in Cenozoic.

Our model has also considered the constraints from distribution of kimberlites which emplaced during Middle Ordovician (470-456 Ma) in the Mengyin (Shandong) and Wafangdian (Liaoning) areas (Liu et al., 2019). Most of previous studies allocated the diamond-bearing kimberlites on both sides of the Tan-Lu fault, with a north-south distance of ca. 550 km between them. Take these two kimberlites as piercing points, we will get a left-lateral displacement of ca. 550 km for the Tan-Lu fault. This magnitude is roughly equivalent to the left-lateral displacement of ca. 540 km estimated by Leech and
Webb (2013), with the correlation constrain of the Dabie and Sulu orogens. However, there are also some diamond-bearing kimberlites in other areas, such as the Huanren, Huludao, and Tieling in Liaoning, and Ji'an in Jilin, in eastern China (Liu et al., 2019). In fact, the distribution of kimberlites in eastern China is oriented in nearly northeast direction, not in the east-west trend (Figs. 3 and 10).

## 5.3 General pattern of back-arc extension

Back-arc extension and breakup of craton block at continental margin are significant manifestation of craton destruction on the Earth surface (Zhu et al., 2020). Previous studies have emphasized the back-arc extension which is roughly perpendicular to the front of the island arc (Ren et al., 2002; Artemieva, 2023). Also a few examples involved local extension parallel to the strike of the trench, especially in the case of oblique subduction, with the development of rift basins and conjugate strike-slip



fault system controlled by normal faulting (Kneller and van Keken, 2008; Balanyáj et al., 2012; Krstekanic et al., 2022). In
this study, we noticed that there are two-direction extensions in the Bohai Sea area during both the Late Cretaceous and
Cenozoic, i.e., the extensions perpendicular and parallel to the subduction zone of the West Pacific plate. Hence, we propose
a general extension model for the back-arc setting, based on spherical geometry consideration (Fig. 12). In this model, the
bidirectional extensions perpendicular and parallel to the subduction zone have the same importance.

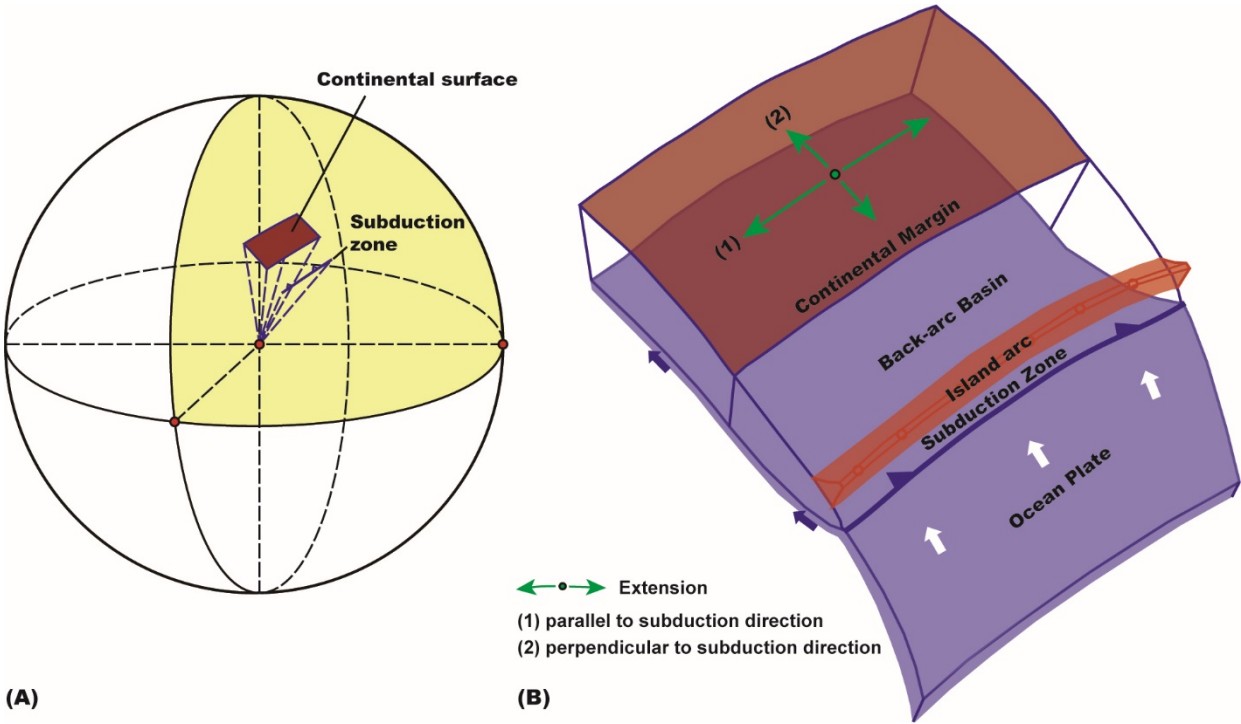

**Figure 12: Proposed model for the back-arc extension in spherical geometry. (A) Spherical geometric diagram of the Earth's surface related to subduction zone. (B) Schematic diagram of bi-directional extension in back-arc continental margin.**

## 6 Conclusions

Based on field investigation, structural analyses, and geological correlation, we constructed a new framework of the fault
system of the Bohai Sea and surrounding area, and reach the following conclusions.

   1. The Bohai Sea fault system is mainly composed of normal faults and strike-slip faults. Superimposed on the rift
system in the Bohai Sea, a left-lateral strike-slip fault formed in the Liaodong Bay in Late Cretaceous and early Paleogene,
while a right-lateral strike-slip fault between the eastern margin of the Liaodong Peninsula and northwestern margin of the
Jiaodong Peninsula formed at the same time. This new mode of movement may have been resulted from the NE stretching
which is parallel to the subduction zone in eastern margin of the Asia Continent.



2. We propose that the formation and evolution of the Bohai Sea fault system is a result from the superimposition of the NE extension parallel to the West Pacific subduction zone on the NW extension perpendicular to the subduction zone. The two-direction extension perpendicular and parallel to the subduction zone should be the basic pattern of the back-arc extension with spherical-geometric effect, especially in the Bohai Sea area.

3. The Tan-Lu fault has at least two-stage evolution, left-lateral strike-slipping in Middle-Late Triassic and Jurassic, and rifting plus left-lateral strike-slipping in Early Cretaceous, respectively. The opening of the Bohai Sea in early Cenozoic has destroyed the previously existing Tan-Lu fault system, resulting in the break-up of the Tan-Lu fault into two segments, the south and north segments, respectively. Both the Honglazi and East Bohai faults are belonging to the north segment of the Tan-Lu fault, while only a few remnants of the Tan-Lu fault remain in the Bohai Sea area.

**Competing interests**

The contact author has declared that none of the authors has any competing interests.

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
