# Peer review of "Origin of the Bohai Sea, North China Craton and implication for bidirectional back-arc extension in East Asia continental margin"

_EGUsphere, 2024_

## Referee Comment (RC3)

[referee-annotated manuscript omitted]

---

## Author Response (AR1)

**Point-by-point Response**

**Reply to RC1**: 'Comment on egusphere-2024-1263', Anonymous Referee #1, 30 May 2024

Comment: The manuscript by Chen and Chen examines the strike-slip faults in East China to talk about the formation of the Bohai Bay basin. Through detailed fieldwork, the authors propose a new two-stage model, namely, the superimposition of the NE extension parallel to the subduction zone on the NW extension perpendicular to the subduction zone. I think it is worth to be published and will attract the attention of the audience in the Solid Earth. Thus, I suggest some revisions before considering accepting for publication.

Meanwhile, I hope my comments and suggestions are useful for your revision.

Reply: Thanks for your review. Your constructive comments and suggestions are very useful for my revision. So, we modified some sentences and photos, and added some materials, to improve our manuscript. The changes are marked in red color.

Comment: In the first paragraph of section 2.1, the authors think of the "Jiaodong and Liaodong Peninsulas" as the key areas. What's your reason? Please clarify it in detail.

Reply: In the first paragraph of section 2.1, we list some reasons to explain why the Jiaodong and Liaodong Peninsulas as the key areas for the formation of Bohai Bay.

Comment: The figure 3 is too large to see the details of structural features in your study area. I hardly find the faults you studied in this map. I suggest that you also provide a detailed structural map of your study area.

Reply: The figure 3 is separated into two maps, a sketched regional map (figure 3) and a detailed structural map of studied area (figure 4), in order to emphasize the structural features of studied area.

Comment: Some photos, such as Fig. 4A, Fig. 5, and Fig. 9 seem not to relate to the topic authors discussed, even presenting as a single figure. I suggest they should be removed. And I think it's better to merge Figure 6 with Figure 7.

Reply: We agree with your comments, some of the pictures do not relate to our topic. We removed these photos, such as Fig. 4A, Fig. 5, and Fig. 9, and merged Figure 6 with Figures 7 and 8.

Comment: There is a logical gap between structure and geodynamics, and I think the bridge is the deformation timing. How do you know the ages of faults you studied, and then correlate them with the opening of the Bohai Bay basin, and even the Paleo-Pacific subduction? I know it is difficult to date the faults, but it's better to give more interpretations and discussions about the deformation timing.

Reply: Really, it is important and difficult to understand the deformation time. We can speculate the faulting time from two aspects. First, previous studies gave us some information about the faulting time. Especially, some of them predicated fault time according to cooling ages in the Luxi area. Second, we have made some logical analysis of fault time. We distinguish fault time according to the relationship between the fault and related strata, especially the effect of the faulting on sedimentary processes. Therefore, we can correlate the normal faulting with the formation of Bohai Bay basin or Bohai Sea. In addition, we can also consider the opening of Bohai Bay basin as the

result of the back arc extension of the Paleo-Pacific subduction, despite there are some gaps between structure analyses and geodynamics, and we need more detailed further studies to reduce the inaccuracy.

Comment: Based on a lot of measurements, Paleo-stress analysis is a useful method to study the kinematics of the faults. I suggest that the authors could add related analysis.
Reply: We have included some preliminary paleo-stress analysis to study the kinematics of the faults, mostly based on previous studies. Further more detailed studies should be conducted in the future.

Comment: The distribution of the magnetic anomaly is a more direct way to discuss the extensional direction, such as the study of the South China Sea (e.g., Barckhausen et al., 2014 Marine and Petroleum Geology). If you can collect the magnetic anomaly data of the Bohai Bay, it may test your two-stage model.
Reply: We have collected data of the magnetic anomaly in the Bohai Sea area, which is modified from Xiong et al., 2015. The data is good evidence for the two-stage model of the studied area.

**Reply to RC2**: 'Comment on egusphere-2024-1263', Chen Wu, 30 May 2024

Comment: All looks great, accepted as suggested.
Reply: Thanks a lot for your kindly review. We have modified the manuscript, in order to improve the writing.

**Reply to RC3**: 'Comment on egusphere-2024-1263', Anonymous Referee #3, 03 Jul 2024
Citation: https://doi.org/10.5194/egusphere-2024-1263-RC3

Comment: This paper deals with the mechanisms of opening of the Bohai sea in NE China. On the formal point of view, the English must be seriously improved, some points are mentioned in the annotated pdf manuscript.
Reply: Thanks a lot for the comments. We have seriously improved our English writing in this time, especially based on the suggestion that the referee mentioned in the annotated pdf manuscript. The modified text has been marked in red.

Comment: From the scientific point of view, there are many flaws in this manuscript.
Reply: Indeed, our manuscript contains some flaws. We have carefully checked the manuscript and earnestly corrected some flaws. Most of the corrections are based on the two referees' suggestion, including another referee.

Comment: (1) In the introduction, the scientific questions must be addressed more clearly as they are. The conclusion must not be presented in the Introduction.
Reply: We have reanalyzed the scientific questions involved in the manuscript, and addressed them more clearly. Also, we have deleted the statements that look like conclusion in the Introduction.

Comment: (2) The Geological Overview section is partial and to some extend incorrect. Many works dealing with the Cretaceous extension, and the MCC in Liaodong and Shandong peninsulas are missing, for instance Lin et al., 2007 GSSP, 2008 J. of Geology, Lin and Wei, IGR; Charles et al., 2012 GR; Qiu et al, 2023 ESR, and many others.

Reply: We have checked the *Regional geological background* section, and prudently present it in current level of understanding, to avoid any incorrect and false statement we can find. However, some geological problems are still not solved in current level, and different views are still needed for the future researches. In this section, we added several references related to MCC in Liaodong and Jiaodong peninsulas, such as Lin et al., 2007 GSSP, 2008 J. of Geology, Lin and Wei, IGR; Charles et al., 2012 GR; Qiu et al, 2023 ESR, and others.

Comment: The term "Indosinian" is improperly used. Collision between N and S China blocks, if any exist east of the Tan-Lu fault, occurred in Early Paleozoic like in the Qinling-Dabie belt. Moreover, "Indosinian" is not a suitable term. This word must be reserved for the S. China-N Vietnam orogen, other orogens of the same age are nor relevant to the same plate convergence system.

Reply: We agree with the referee's suggestion that the term "Indosinian" should be reserved for the S. China-N Vietnam orogen. To eliminate misconception, we have modified the using of term "Indosinian", instead as "Triassic". We added some references on the collision between N and S China blocks, also Faure et al., 2014 on the S China block-Indochina collision. Many geologists from China and other countries may have different views than the referee on the formation time of Qinling-Dabie-Sulu orogenic belt, which is also named as the Central China Orogenic Belt. They thought that the Qinling-Dabie-Sulu belt was formed in the Triassic, resulting from the collision between N and S China blocks in that time. They believe that there is still the existence of the Qinling Ocean between the N and S China blocks in the Late Permian and Early Triassic. Many researchers considered the Tan-Lu fault as a transform fault which offset the Qinling-Dabie and Sulu belts.

Comment: (1) Fig. 3 is too complex and thus not understandable. The geological map of this area must be redrawn in order to highlight the important features.

Reply: We have simplified and redrawn Fig. 3, to emphasis some important tectonic features in this area. At the meantime, we added Fig. 4 to focus the studied area around the Bohai Sea. We have rearranged the figure numbers.

Comment: (2) Often the structural field evidence for normal and strike-slip faults are not convincingly provided. Figure 4 presents complex structures that are hardly believable. The differences between the red and pink lines are not mentioned. Fig 4A is useless.

Reply: We accepted the referee's suggestion, deleted the previous Fig 4A, and rearranged the other figures. The structural relationship drawn in the figures is the result of what we observed on the outcrop. Their geometry and kinematics are believable. To avoid confusion, we added some original photos as comparisons with explained structural results. Also, we added some legends and explanations for different colored faults in the figure and caption.

Comment: In section 3.2, there is a mess between the brittle joints, the faults, and the magmatic

structures observed in the Cretaceous granites. As a whole, the kinematics of the strike slip faults is not convincingly presented too.

Reply: Thanks for the referee's remind. However, we don't think there is a mess. The brittle joints, faults, and magmatic structures are developed during different stages and different strain fields, which represent different stress statues with different kinematic implication. They reflected the complexity of fault activities and structural developing. In the revised manuscript, we have given further explanations of fault kinematics, although some branch faults have very little displacement. We believe that the arrangement of extensional brittle joints could be adopted as kinematic indicator of related strike-slip faulting.

Comment: Section 4 is hard to understand. The authors seem to argue that due to their geological similarities the Jiaodong, Liaodong and Korea peninsula experienced strike-slip and normal faulting. Even if the 3 areas have obviously common geological features since Archean to Cretaceous at least, these similarities do not prove at all strike slip or normal offset.

Reply: In the revised manuscript, we added evidence from aeromagnetic anomaly, to explain the rationality of the block comparison surrounding the Bohai Sea. We know that, for most professional researchers, our model is hard to understand. This is because, most researchers have already had pre-existing or solidified traditional understanding of regional geology and tectonics. They may start off from the aspect of the Tan-Lu fault, believed that the north extending of Tan-Lu fault may have no problem, and the fault activity is not too complex. However, if we stand from the perspective of Bohai Sea, we will find that, it is hard to track the footprints of Tan-Lu fault within the Bohai Sea area, although there is only 20-meter depth of the sea water. The aeromagnetic anomaly of Bohai Sea area shows that the Archean basement of Liaodong Peninsula can be connected with the basement in the Laizhou Bay, rather than the Archean basement of Jiaodong Peninsula. We conceive that different faults had their different activities in different periods, with different natures. Although some faults may have almost the same location in the surface, they cannot be considered as the same fault, since they have different natures in different periods and different extending in the depth. Therefore, we suggested a tectonic model which is different from previous studies, to solve the complicated superimposed tectonics in the Bohai Sea area.

Although there may still exist some questions and flaws in our model, however, we think that it has made some significant improvements compared to previous studies, especially on the Tan-Lu fault. About the normal faulting, our understanding is basically derived from a series of regional geological surveys, marine geological surveys, and oil and gas geological surveys made by many geological researchers in China. Their achievements on normal faults are mainly reflected in the Geological Map of Asia (Ren et al., 2013) and Geological Map of China (China Geological Survey, 2004), as well as a series papers on regional oil and gas exploration in the Bohai Sea area. In the process of oil and gas exploration, some understanding of strike-slip faults has also been proposed, mainly concentrated in the Liaodong Bay area. In our manuscript, we propose a new perspective on the understanding of JLF, the Jiao-Liao fault, as a right-lateral strike-slip fault, offsetting the Jiaodong and Liaodong Peninsulas. The current understanding of Tan-Lu and Jiao-Liao faults is still relatively superficial and simple. We look forward to more work that can validate, improve, or even overthrow our model.

Comment: Fig 11 is not understandable. The genetic model of the Bohai Sea requires additional explanations.

Reply: Based on the referee's suggestion, we made further modification of the previous Fig. 11 (now Fig. 9), and added more words explanation for this figure in the text.

Comment: In conclusion, I cannot recommend the publication of this manuscript in its present form. More work is necessary to present convincing data on the fault kinematics. The text and figure must be also improved in order to built up an understandable geodynamic model.

Reply: We appreciate all the constructive suggestions from the referee, which allowed us to do more carefully check of the work in the manuscript. We think that our current work can basically support our current tectonic model for the formation of Bohai Sea. We expect that more work should be done in the future and more researchers should be involved in the discussion. In this time, we have made serious modifications of the text and figures, to improve the manuscript. Thanks again for the referee's review and constructive suggestions.

---

## Author Response (AR2)

**Point-by-point Response**

**Reply to:**
09 Oct 2024, Topic editor decision: Publish subject to minor revisions (review by editor), by Yang Chu

**Comment**: Public justification (visible to the public if the article is accepted and published):
One of the previous reviewers and myself have both finished reviewing and give some new comments on this manuscript. Although the review suggests minor revision, but more work should be done before its final acceptance. As the reviewer points out, the current structural evidence cannot support the tectonic model. This manuscript staddles between a review and a research paper, providing only a large-scale geological map and field photos, while lack of small-scale geological maps of the region and cross-cutting relationship hinders reconstructing deformation sequence. By adding new geological maps and field evidence, this manuscript can create a solid link from deformation features and superposition to deformation sequence and their tectonic interpretation.
**Reply**: Thanks for review. Though there are many deficiencies in our MS, we still think that our model is much better than previous studies. The improvements includes the re-interpretation of the Tan-Lu fault zone, and the model for the formation of the Bohai Sea Basin. Although the Bohai Sea Basin is a part of the Bohai Bay Basin, there are still many differences between them. Therefore, we focused our discussion on the Bohai Sea Basin, not the Bohai Bay Basin.

**Comment**: I totally agree with the reviewer's third point, which also bothers me a lot, that is the mixing of geological and geographical terms. For example, Bohai sea is a geographical unit, which covers half area of the Bohai Bay Basin, and the boundary faults have nothing to do with, or plays a negligible role in the formation of the Bohai sea. Jiaodong and Liaodong peninsulas are also confusing. If they are irrelevant to tectonic blocks, just abandon them after the geological setting. In the sections of Introduction and Geological setting, such mixed-use leads to different understanding and interpretation in the final discussion. I strongly suggest removing all of geographic terms, except those for locating geological units. In addition, every local names must be marked on maps to locate.
**Reply**: Thanks. We have changed the Bohai Sea into the Bohai Sea Basin or Bohai Sea region. Also, we changed the Jiaodong and Liaodong peninsulas into the Jiaodong and Liaodong blocks.

**Comment**: Deformation of faults and the study region also lacks one section to list field and chronological evidence before discussing the structural evolution, i.e. the part 4.3.
**Reply**: In this MS, we focused the discussion on the Tan-Lu fault, which is the most important fault related to our model for the origin of the Bohai Sea Basin. Since there are so many faults in the studied area, we have no place to discuss all of them.

**Comment**: Given these problems, another round of revision is necessary. Although there may be a lot of corrections, the main part does not need a substantial change, and I think this new idea will attract more discussion in the geological evolution of the Bohai Bay Basin. Here I give a recommendation of moderate revision, but the authors should be careful when incorporating the comments into the revision of text.

**Reply**: Thank you very much for the review. We will make careful revisions and hope our modifications can make significant improvements of the MS.

**Comment**: Some changes in text and figures:

Title: Bohai Bay Basin is better than Bohai Sea.

**Reply**: Thanks. We think this MS is about the origin of the Bohai Sea Basin, not the Bohai Bay Basin, so, we changed the title as *Origin of the Bohai Sea Basin, North China Craton and implication for bi-directional back-arc extension in East Asia continental margin*.

**Comment**: Line 7: northwestward

**Reply**: Okay.

**Comment**: Line 15: correlation->comparison

**Reply**: Okay.

**Comment**: Line 19: trench-parallel and trench-perpendicular extension

**Reply**: Okay.

**Comment**: Line 24-34: This is what I am baffled. Origin of a sea may have many climatic and oceanographical reasons, much different from origin of a basin.

**Reply**: Thank you. We have changed the Bohai Sea into the Bohai Sea basin or Bohai Sea region.

**Comment**: Line 29-30: What is this region? Most large earthquakes occurred onshore.

**Reply**: We have changed it into the BBB and surrounding region.

**Comment**: Line 31: deconstruction->destruction

**Reply**: Okay.

**Comment**: Line 56: Strange to have geographical and geological units together.

**Reply**: We have changed the Bohai Sea into the Bohai Sea region.

**Comment**: Line 70: Same as the last comment.

**Reply**: We have changed the Bohai Sea into the Bohai Sea region.

**Comment**: Line 81-84: These sentences have nothing to do with the geology.

**Reply**: Okay. We have deleted these sentences.

**Comment**: Line 85: Jiaodong Block is more important than Jiaodong peninsula in this ms.

**Reply**: Yes. We have changed the Jiaodong peninsula into the Jiaodong Block.

**Comment**: Line 98: with similar time to->coeval

**Reply**: Okay.

**Comment**: Line 121: Late Jurassic continental arc

**Reply**: Thanks. We have rewritten the sentence.

**Comment**: Line 122: Why is the Shizhuzi pluton emphasized here? Where is it?

**Reply**: The Shizhuzi pluton is a small pluton in the Liaodong block. It is not important. We have deleted it.

**Comment**: Line 145: Meaningless sentence. Use geological units in this paragraph!

**Reply**: Okay.

**Comment**: Line 169: Strike-slip faulting

**Reply**: Okay.

**Comment**: Line 187: Delete much later stage

**Reply**: Okay.

**Comment**: Line 189-191: Awkward sentence. Consider rephrasing.

**Reply**: Okay. We rephrase the sentence.

**Comment**: Line 193: This fault is suspicious. Could you provide some direct evidence for the fault, instead of dashed lines in the figure?

**Reply**: The Honglazi fault is a coastal fault. It is inferred from the different distribution of strata layers in its both sides. In its western side, there are pre-Cenozoic strata outcropped in the surface. However, it is difficult to know what strata are in its eastern side in the Bohai Sea, except for possible Cenozoic sediments. Therefore, there are some kind tectonic relief between the two sides of the fault.

**Comment**: Line 207-208: I need a detailed map to see it.

**Reply**: The EBF is a coastal fault. The relationship between both sides of the fault is clear. More detailed map is not available in this time.

**Comment**: Line 240: weathered paleo-crust. One-meter-thick magenta clay layer.

**Reply**: Okay. Thanks.

**Comment**: Line 246: What do you mean the North Yellow Sea fault form the Jiao-Liao fault?

**Reply**: We change the form into compose. What we mean is that the North Yellow Sea fault is a segment of the Jiao-Liao fault.

**Comment**: Line 278: Again, peninsula and block are together.

**Reply**: Okay. We change the peninsulas into blocks.

**Comment**: Line 289: interrelated->coeval. Delete just

**Reply**: Okay.

**Comment**: Line 296: gold deposits

**Reply**: Okay.

**Comment**: Line 307: delete an assumption of
**Reply**: Okay.

**Comment**: Line 310: Bay may not be opened, but basin is.
**Reply**: Okay.

**Comment**: Line 314: Change to NE-striking extensional structures and NW-SE extension, and Cenozoic WNW-striking normal faults.
**Reply**: Okay.

**Comment**: Line 316: Delete this is to say. Common->similar
**Reply**: Okay.

**Comment**: Line 320-321: Better discuss the geological connection, not geographical connection.
**Reply**: You are right. Previous studies have detailed correlational analyses between the Jiaodong block with northern part of the Korean peninsula, which makes it clear that there are something losing in the North Yellow Sea Basin. However, we do not know very well what are the really loser in the North Yellow Sea. More detailed geological and geophysical correlation is still needed for the future.

**Comment**: Line 324: correlational analysis->comparison
**Reply**: Okay.

**Comment**: Line 328: Is there any age constraint on the Jiaoliao, East Bohai and Honglazi Faults?
**Reply**: Currently, there is only geological constraint on faulting time of the Jiaoliao, East Bohai and Honglazi Faults. In the future, we still need to do some geochronology work to make more detailed constraints on the faulting times.

**Comment**: Line 344: It is not useful to have several sentences the describe location of the segments of the Tanlu Fault. Mark the different segments on map.
**Reply**: Okay.

**Comment**: Line 351: Bohai bay…
**Reply**: Okay.

Comment: Line 354: Delete other.
**Reply**: Okay.

**Comment**: Line 364: Delete Anhui and Jiangsu provinces.
**Reply**: Okay. (Line 366)

**Comment**: Line 377: Show Dongying depression on the map.
**Reply**: Okay.

**Comment**: Line 378: Delete that is to say.
**Reply**: Okay.

**Comment**: Line 378-379: Not a clear sentence.
**Reply**: The sentence is rewritten as a clear one.

**Comment**: Line 381: excluded as a part of…
**Reply**: Okay.

**Comment**: Line 383: What method has been used for these ages?
**Reply**: Apatite FT dating of footwall rocks of extensional normal faults.

**Comment**: Line 385: Use widespread for universally developed.
**Reply**: Okay.

**Comment**: Line 391: Delete magnitudes.
**Reply**: Okay.

**Comment**: Line 393: How is 340 km calculated?
**Reply**: The displacement of ca. 340 km is calculated from the model, when the Jiaodong and Liaodong blocks are put together, and the Jidong and Liaodong blocks are also put together.

**Comment**: I do not find any new implications and discussion in section 5.3. All the sentences and figure 11 can be deleted.
**Reply**: We deleted the figure 11 and all the sentences in section 5.3.

**Comment**: Figure 1: Nice figure. What do different colors represent? Show the location of the Bohai Sea and BBB.
**Reply**: Thank you. We have shown the tectonic regimes represented by different colors in the figure. Due to that the map belongs to the entire Asian region, the Bohai Sea and Bohai Bay Basin are relatively small and not suitable for differentiation. Therefore, figure 1 only shows the approximate location of Bohai Sea (BS). The specific ranges of Bohai Sea and BBB are shown in Figure 3.

**Comment**: Figure 3: What is the evidence for the block boundary? Fault? Surface outcrop areas?
**Reply**: We divide the scope of geological blocks based on mainly the distribution of faults and geological bodies. These blocks were relatively independent in the Cenozoic, but were mostly connected in the pre-Cenozoic.

**Comment**: Figure 5: Difficult to read the text with red colors. We have an original pic for 5C, why not one for 5A?
**Reply**: We have changed most of the red characters in the picture to white characters for easier recognition. Due to the large space occupied by the images, we only provided the original image of Figure 5B. The local structure in Figure 5B is much detailed, while Figure 5A is more regional.

**Comment**: Figure 6: Same problem as Figure 5.

**Reply**: We have changed most of the red characters in the picture to black ones for easier recognition. Due to the large space occupied by the image, we only provided the original image of Figure 6B. The information in this image is more typical and detailed.

**Comment**: Figure 7A: I do not see early fractures cut by later ones. Any clear photos?

**Reply**: Both the early and late fractures belong to the nature of fractures and do not have significant displacement, the cutting relationship between them is not particularly obvious. In fact, they formed almost simultaneously and cut each other apart. However, early developmental fractures have a longer duration and length.

**Comment**: Figure 9: Any geological evidence for the 162 km displacement?

**Reply**: The current geological evidence related to the displacement between the Jiaodong and Liaodong peninsulas mainly comes from the comparison of Early Cretaceous extensional structures (such as metamorphic core complexes) and granite intrusive rocks between the two peninsulas, as well as the distribution of pre-Cretaceous geological bodies and the development of fault structures. The Linglong dome has a good connection with the Liaoning metamorphic core complex, which can be used to calculate displacement, approximately 162 km.

---

## Author Response (AR3)

**Point-by-point Response**

**Reply to:**
29 Oct 2024, Topic editor decision: Publish subject to minor revisions (review by editor), by Yang Chu

**Comment**: I only have several comments. One is that I don't find replies to the comments from the reviewer.
**Reply**: Thanks for the comments. We have tried to find the comments from the last reviewer, and compensate the reply.

**Comment**: Second, define the Bohai sea basin. Bohaibay basin has its tectonic meaning, but the bohai sea basin doesn't. What are the strata and boundaries of this new basin? Please give the details to show the necessity of this new basin.
**Reply**: Okay. We have defined the Bohai sea basin in the text. This is not a newly defined basin. The Bohai sea basin is defined as a Cenozoic basin bounded by the Jiaodong Peninsula, North China Plain, Yanshan Mountains, Liaohe Bay Plain and Liaodong Peninsula. It is mainly composed of the Bohai Sea area. We have added one reference for this (Wang, T., Pan, X., and Hao, F.: Hydrocarbon accumulation and distribution prediction in Bohai Basin [in Chinese], Beijing, China, Geological Publishing House, 1–492, 2016).

**Comment**: Third, Delete figure 5c and 6c because they are not useful to show any additional information.
**Reply**: Okay. We have deleted figures 5c and 6c.

**Comment**: Lastly, delete the block boundary lines in figure 3. Sedimentary contact cannot define the boundary between blocks. For example, we never attribute the boundary of the North China Craton to the boundary of the Cenozoic Bohai bay basin.
**Reply**: Okay. We have deleted the block boundary lines in figure 3.

**Compensating reply to the last reviewer**:
**Report #1**: (Submitted on 13 Sep 2024, Anonymous referee #1)
1) Scientific significance, Good.
2) Scientific quality, Good.
3) Presentation quality, Good.
For final publication, the manuscript should be accepted subject to: minor revision.
Were a revised manuscript to be sent for another round of reviews: I would not be willing to review the revised manuscript.
Suggestions for revision or reasons for rejection: Minor revision.
**Reply**: Thank you very much for the review. We will make careful revision for the MS.

---

## Author Response (AR4)

**Point-by-point Response**

**Reply to:**

07 Nov 2024, Topic editor decision: Publish subject to minor revisions (review by editor), by Yang Chu

**Comment**: Please find the attached comments from reviewer 1, and reply and make changes to them. I cannot accept it if this procedure is incomplete. This is important!

**Reply**: Okay. We carefully studied the comments from reviewer 1, and revised the manuscript accordingly. Thank you.

**Reply to the reviewer 1**:

**Comment**: After the first round of revisions, the manuscript has shown improvements in both the text and figures. The authors have carefully addressed the reviewers' comments and revised the manuscript accordingly. However, I have identified some remaining issues that require minor revisions.

**Reply**: Thank you very much for the comments from reviewer 1.

**Comment**: 1. A gap persists between the structural data and the tectonic model. In the last paragraph of Section 3.1, it is mentioned that the Honglazi area experienced extension, left-lateral strike-slip, followed by extension. It remains unclear how these deformations are integrated into your three-stage model. I suggest adding a few sentences to clarify how these deformation phases are incorporated into the model.

**Reply**: Thank you very much for the suggestion. We have added some explanations in section 4.3, to incorporate the deformation phases from observation into the model.

**Comment**: 2. Lack of robust argumentation in the discussion section. For example, in Sections 4.1 to 4.3, the three-stage model is largely based on comparisons of geological events in the Jiaodong, Liaodong, and Jidong areas, rather than on your new structural data. Particularly, do you have any other direct evidence to support that the Jiao-Liao fault is connected to the Yalvjiang fault, such as geophysical data? Additionally, the discussion about the Tan-Lu fault is somewhat unclear, which seems to be a significant scientific issue that extends beyond the scope of your current data.

**Reply**: We have made serious revisions to the discussion section. About the relationship among the Jiao-Liao fault, NYSF, and Yalvjiang fault, we have supplemented explanation evidence from the aeromagnetic anomaly map. Due to the numerous issues with the Tan-Lu Fault, we only discussed its relationship with the Bohai Sea Basin. More data are needed for the future discussion on the Tan-Lu fault system.

**Comment**: 3. Excessive use of local geographical names, which may be difficult for international readers to understand. Terms such as Bohai Sea, Bohai Bay Basin, North China Plain, Lower Liaohe Plain, and various city names may pose challenges for foreign readers. Furthermore, "Bohai Sea" and

"Plain" seem to be geographical terms rather than geological ones.

**Reply**: Such is the case. We have tried to avoid using too many geographical terms. We have already used the Bohai Sea Basin to replace the region where the Bohai Sea is located. Although the North China Basin and the Lower Liaohe Basin can also be used to replace the North China Plain and Lower Liaohe Plain, respectively, as these two basins are not the focus of this study, we think it is better not to replace them. As for local city names, we have made significant deletions in this revision and believe that it is necessary to retain a certain amount of city names.

**Comment**: Some minor issues:

Figures 4, and 8: In these two figures, there are many inferred faults shown beneath the Bohai Sea and Cenozoic cover. Do you have some evidence, or are they sourced from some references?

**Reply**: We have reviewed the fault system in figures 4, 7, and 8 carefully. Among the faults, the Honglazi fault (HF) and East Bohai fault (EBF) are the faults inferred from our field observation in this manuscript. They developed along the coast and underwater of the Bohai Sea, with evidence from outcrop analysis. The Jiao-Liao fault (JLF) is inferred mainly from aeromagnetic anomaly map in figure 7. Other faults underwater in the Bohai Sea Basin are all from literatures such as Allen et al., 1997, Ren et al., 2013, and Wang et al., 2016. The relevant explanations have been added for the figures and in the main text.

**Comment**: Lines 63-64: Repetitive

**Reply**: Okay. We merged the sentences and eliminated the repetitive expression.

**Comment**: Lines 70-73: Regarding the linkage of the Tanlu fault and the Bohai Sea, it needs to be clear the significance of the Tanlu fault to understand the formation of the Bohai Sea.

**Reply**: Yes. Previous studies have rarely involved the relationship between the Tan-Lu fault and Bohai Sea Basin. They always inferred simply that the Tan-Lu fault as the eastern boundary of the Bohai Bay Basin. However, there is no conclusive evidence to determine whether the Tan-Lu fault exists in the Bohai Sea Basin and whether it constitutes the eastern boundary of the Bohai Sea Basin or Bohai Bay Basin.

**Comment**: Line 129: I don't think the Yanshan is an orogenic belt.

**Reply**: We think you are right. The Yanshan Mountains are not an orogenic belt, but a fold and thrust belt formed in late Mesozoic. Therefore, we replaced the Yanshan orogenic belt with the Yanshan fold-thrust belt.

**Comment**: Line 210: Explain how extensional fractures relate to left-lateral slip striking.

**Reply**: Okay. According to the relationship between the extensional fractures and fault motion (or shearing) direction, the acute angle between extensional fractures and shearing surface indicates the motion direction of the standing wall. Appling this criterion, we inferred that the early fractures in figure 6A indicate left-lateral slip striking.

**Comment**: Line 214: What's the meaning of the "general"? Sub-horizontal? Gentle?

**Reply**: We are sorry. The "general" is the wrong expression of "gentle". We use sub-horizontal to replace the general in this time.

**Comment**: Figures 5 and 6 can merged into one and removed 5c and 6c.
**Reply**: Okay. We have already adopted this modification suggestion in the previous revision, and merged the previous figures 5 and 6 into figure 5.

**Comment**: Line 231: Why is a strike-slip fault horizontal?
**Reply**: We have made modifications, to remove the "horizonal". We are not saying that there is a horizontal strike-slip faulting, but rather, the horizontal striations on vertical surface indicate a movement pattern consistent with the strike-slip faulting. That is to say, the faulting plane of the strike-slip fault is almost vertical.

**Comment**: Line 280: Jiao-Liao-Ji tectonic belt.
**Reply**: Okay. We modified the Jiao-Liao-Jilin tectonic belt into the Jiao-Liao-Ji tectonic belt.

**Comment**: Line 286: in→into
**Reply**: Okay.

**Comment**: Line 313: Deleted "northern"
**Reply**: Okay.

**Comment**: Line 315: What's the meaning of "extension in NW"? NW-SE extensional tectonics?
**Reply**: The meaning of "extension in NW" is "extension in NW direction", which is equal to the NW-SE extensional tectonics.

**Comment**: Line 326: Add the approximate age ranges for each stage to offer more context and detail.
**Reply**: Okay. We added the approximate age ranges for each stage.

**Comment**: Line 329: Most authors suggest NW-SE extension (perpendicular to the subduction zone) dominated during the Early Cretaceous, and please explain how NE-SW extension (parallel to the subduction zone) fits into the tectonic framework.
**Reply**: In the Yanshan fold-thrust belt located on northwest side of the Bohai Sea Basin (as shown as the Jidong Block in this manuscript) and the Jiaodong Peninsula on southeast side of the Bohai Sea Basin, there are indeed NW-SE extensional deformation mainly in the late Early Cretaceous. However, in the Bohai Sea Basin, especially in the Liaodong Bay area, the NW-SE extension mainly occurred in the Cenozoic (figure 4), controlling the development of the graben basin in this region. The NE-SW extension in late Early Cretaceous mainly occurred in the Luxi (western Shandong) region (figure 3). At the same time, the Yi-Shu Fault Zone, which is always considered as a part of the Tan-Lu Fault Zone, developed the NNE-trending rift system, indicating the WNW extension in late Early Cretaceous. This is to say, the extensional activity in late Early Cretaceous is not limited to the NW-SE direction, but also

the NE-SW and WNW directions. The Cenozoic regional structural framework in the Bohai Sea Basin is mainly controlled by NW-SE, NE-SW, and nearly N-S extensions.

**Comment**: Line 343: The discussion about the Tan-Lu fault is beyond your data.
**Reply**: We believe that it is necessary to discuss the evolution of Tan-Lu Fault, since this is directly related to the origin of the Bohai Sea Basin. Previous studies mostly believed that the formation and evolution of the Bohai Bay Basin and Bohai Sea Basin were controlled by the Tan-Lu Fault, and the Tan-Lu Fault constitutes the eastern boundary of the Bohai Bay Basin. The aeromagnetic anomaly map (figure 7) shows that the Tan-Lu Fault cannot be connected to the Liaodong Bay and Yilan-Yitong Fault (YYF) on the north side through the Bohai Sea Basin. Instead, the Tan-Lu Fault is modified by the extensional structures in the Bohai Sea Basin. Therefore, we need to discuss the Tan-Lu Fault and modify our understanding of it, when we are talking on the genesis of the Bohai Sea Basin. In our model, we believe that the HF and EBF belong to the same fault zone as the left lateral strike-slip Tan-Lu fault, during late Early Cretaceous to early Cenozoic. The HF and EBF are separated only in Cenozoic due to NW extension in the Liaodong Bay. According to previous studies, we believe that the dominated Cenozoic right lateral strike-slip faults developed in the Bohai Bay Basin and Bohai Sea Basin, are not parts of the left lateral strike-slip Tan-Lu fault system.

**Comment**: Line 414: What's the evidence of the extension parallel to the subduction zone?
**Reply**: The evidence for extension parallel to the subduction zone mainly comes from: 1) the NE extension during the late Cretaceous in the Luxi (western Shandong) region (figures 3, 4, and 7), which is nearly parallel to the NE trending subduction zone on the northwest side of the Philippine Sea Plate; and 2) the basement extension along the NE direction in the central Bohai Sea and northern Yellow Sea areas, in the aeromagnetic anomaly map (figure 7), which is also nearly parallel to the NE trending subduction zone on the northwest side of the Philippine Sea Plate. This part of description has been added to the section 3.4.